# Development of a Matrix Based Statistical Framework to Compute Weight for Composite Hazards, Vulnerability and Risk Assessments

**Rubaiya Kabir [1],\*, Marin Akter [2], Dewan Sadia Karim [1], Anisul Haque [1], Munsur Rahman [1] and Mohiuddin Sakib [1]** 

[1] Institute of Water and Flood Management, Bangladesh University of Engineering and Technology, Dhaka-1000, Banghladesh; dewan.sadia123@gmail.com (D.S.K.); anisul@iwfm.buet.ac.bd (A.H.); mmrahman@iwfm.buet.ac.bd (M.R.); sakib.mohiuddin06@gmail.com (M.S.)

[2] Department of Mathematics, Bangladesh University of Engineering and Technology, Dhaka-1000, Bangladesh; marin.akter12@gmail.com

\* Correspondence: orchi36@gmail.com; Tel.: +88-(0)-1674-894916

**Abstract:** Selection of relative weights for different indicators is a critical step during assessment of composite hazards, vulnerability, and risk. While assigning weight to an indicator, it is important to consider the influence of an individual indicator on a particular composite index. In general, the larger the weight of the indicator, the higher the importance of that indicator compared to other indicators. In this study, a new matrix based statistical framework (MSF) for weight assignment is developed that can be considered as the simplest and most accurate method for assigning weights for a large number of indicators. This method (MSF) is based on the valuation of the correlation matrix and Eigenvector associated with Eigenvalue. Relying on the inter build up methodology, MSF can fulfill some built-in gaps among other weightage methods. It can also directly give the 'decision' to select the relative weights that are found from the Eigenvector corresponding to the largest Eigenvalue. The new method is applied by assigning weights to 15 socio-economic indicators and assessed vulnerability and risk in the Bangladesh coast. While comparing with other weight methods, it is found that MSF gives the most acceptable physical explanation about the relative values of weights of indicators. In terms of accuracy, MSF is found to be most accurate compared to other weight methods. When large numbers of indicators are involved in an application, MSF is found to be relatively simple and easy to apply compared to other methods.

**Keywords:** vulnerability; risk; storm surge hazard; indicators; weight; Eigenvalue and Eigenvector; matrix; Bangladesh

## 1. Introduction and Statement of Problem

Responding to climate change is now widely acknowledged as one of the greatest challenges facing society [1–3]. Every day new issues related to socio-economic factors and ongoing climate change impacts are emerging. To address this problem and related concerns, hazard, vulnerability, and risk assessments are required that can help to understand the complex set of factors that contribute to the assessment of how communities will adapt to changing environmental conditions. Currently, integrated assessments focusing on climatic hazard specific risk minimization on coastal deltas are rare [4–7]. As a result, vulnerability and risk in coastal deltas are not fully understood and the identification of risk reduction and adaptation strategies following appropriate ranking of indicators are often based on incomplete assumptions.

Computing hazard, vulnerability, and risk assessments that combine physical and socio-economic information to show climate change risks in a sector have become a way to address the need for minimizing climatic risks [8,9]. There are several studies on quantifying and comparing vulnerability in terms of an index [10]. Composite indices of vulnerability are analytical, communicative, and collaborative tools, which help to raise awareness, support decision making, and facilitate planning and policy development through improved understanding of a complex multidimensional problem [11]. In vulnerability assessments, indicators are used to 'measure' and 'characterize' the vulnerability of a system. Vulnerability depends critically on the indicators that make a system vulnerable to a hazard. It has been well recognized that appropriate weight selection for indicators has important strategic implications for the assessments to be spatialized, because a specific decision normally will involve long-term commitment of resources and be irreversible in nature. Hence, establishing an unbiased (in terms of human perception), universally applicable (in terms of geographical location), and simple (in terms of application) method will play a great role in this regard.

*Research Gap, Research Question, Objectives, and Significance*

During the assigning of rankings of indicators in composite hazard, vulnerability, and risk assessment, computation of relative weight is a very important step. As all measured indicators do not contribute with equal importance, the weighting approach needs a proper conceptual definition. The ideal weighting method for the computation of hazard, vulnerability, and risk assessment should be transparent, with accurate and proper valuations, and needs to produce comparable index values according to spatial and temporal variations of hazards. The key notes that need to be considered are: (i) the characteristics and the rationale of the indicators, (ii) the meaning and the contribution of the indicators, and (iii) the data quality and the statistical adequacy of indicators [12]. All these aspects are not highly focused upon in relevant literatures. There exists a missing link as all the methods suggested by different researchers are either biased or context specific. There are various practices that are currently used for weight assignment. However, all of them have some inbuilt gaps. Furthermore, there are large gaps in the literature regarding identification of a best suitable method for giving weights to the indicators. Some research questions raised from different literatures include: (a) what is the process of prioritizing the variables from different set of indicators? (b) how to determine relative weights from the Eigenvector corresponding to the highest Eigenvalue? (c) how the prescribed method in any study can be applicable for other geographical settings?

To answer these research questions, the main objective of this paper is to introduce, describe, explain, and apply a new method for assigning weights to a large number of indicators. The new method is called the 'matrix based statistical framework' (MSF). The specific objectives of the present paper are:

(a)　To form the correlation matrix between different sets of indicators.
(b)　To determine relative weight from the Eigenvector corresponding to the highest Eigenvalue.
(c)　To show applicability of this method for giving appropriate weights to the indicators.
(d)　To test the validity of this method by assessing vulnerability for the Bangladesh coast.

This paper focuses on filling the gaps of current weight computation methods prescribed by different researchers that fail to address the interdependencies of indicators in both explicit and implicit ways. This work aims to compute the relative weights of a set of indicators, which are not based on human perception or geographical context. The significance of this paper is that due to its simplicity and easy-to-apply methodology (a pseudo code of the method is provided at the end of the paper as an appendix), users can easily apply this method for prioritizing the most important indicators by giving relative weights to large number of indicators in any geographical setting.

## 2. Overview of Other Weightage Methods and a Comparison with the New MSF Method

Giving weight based on someone's opinion may vary person to person. For example, the analytical hierarchal process (AHP) method cannot be satisfactorily applied to weight requirements because human judgment cannot be quantified. Imposing scale for judgments creates inconsistencies because of the judgment of the individual decision maker. As AHP depends on the capacity and background of the decision maker, it creates inconsistencies on the judgment. It is also true for the expert weighting method as expert weighting may not be transferable from one area to another. For weighting assignment, the potential problem arises in building up an expert panel or group. Again, allocating a certain weight over too large a number of indicators may lead to a state of serious agitation for the experts as it implies uncertain or confused thinking. Furthermore, when experts give weights to indicators, they generally vary from expert to expert depending on their specialization. In case of an equal weighting method there exist no statistics or empirical grounds for choosing the values of the weights and the results reflect insufficient knowledge of causal relationships. With equal weighting, no distinction can be made between the relative or absolute valuation of indicators. If we consider the survey weighting method, this weighting method requires a large number of comparisons for validation and it is computationally expensive. It can produce inconsistency when dealing with many indicators. It basically depends on the sample of respondents chosen and on how questions are framed in specific surveys. This approach downplays the relative importance of local stakeholder knowledge for weighting design, which is a massive oversight to vulnerability and risk mapping. Like all other social survey methods, PRA (Participatory Rural Appraisal) is highly biased by human perception. Many of the above mentioned methods are beset with the bias issue except principal component analysis (PCA). Yet PCA has some gaps that can be fulfilled by the MSF method. Principal components are selected based on explaining the maximum variance present, which mathematically reduces the most important indicators in general. The relative weights of indicators produced by the PCA technique are unable to give the importance to the impact of an individual indicator.

On the contrary, the MSF method is applicable for any region and any type of scientific research for giving relative weights to any set of indicators. This new method is an effective way to get a simplified and interacting way of weighting for composite hazards, vulnerability, and risk assessments for any region.

## 3. Advantages and Disadvantages of MSF

Advantages of MSF:

- MSF does not need to consider how many Eigenvalues are greater than 1; it considers just what is the largest Eigenvalue, which implies the Eigenvector is considered as the relative weights of the variables.
- MSF directly gives the 'decision' to select the Eigenvector as relative weights that corresponds to the largest Eigenvalue.
- In terms of application to assign weights to large number of indicators, MSF is comparatively easy and simple to apply compared to other methods. It does not need any 'decision' to be taken about the 'components'.
- MSF has one component vector; in that case, the expected relative weights of indicators does not depend on the linear combination of component vectors with the variances (weights of component vectors), which is simple.

Disadvantages of MSF:

- MSF considers linear assumptions between the variables during the computation of Eigenvectors corresponding to Eigenvalues.
- In MSF, there are many statistical distributions where mean and covariance do not give relevant physical information of variables.

## 4. Methods

MSF was established on the basis of the Eigenvector corresponding to the maximum Eigenvalue from the correlation matrix, which was constructed from Pearson correlation coefficients. Pearson correlation coefficients were established from the summation of squared matrix, which is the relationship between the variables to each other. Pearson correlation coefficients are mathematically defined as:

$$\left.\begin{aligned}
P_{xy} &= \frac{SS_{xy}}{\sqrt{(SS_{xx}+SS_{yy})}} \\
P_{yz} &= \frac{SS_{yz}}{\sqrt{(SS_{zz}+SS_{yy})}} \\
P_{xz} &= \frac{SS_{xz}}{\sqrt{(SS_{xx}+SS_{zz})}} \\
P_{yx} &= \frac{SS_{yx}}{\sqrt{(SS_{xx}+SS_{yy})}} \\
P_{zy} &= \frac{SS_{zy}}{\sqrt{(SS_{zz}+SS_{yy})}} \\
P_{zx} &= \frac{SS_{zx}}{\sqrt{(SS_{xx}+SS_{zz})}}
\end{aligned}\right\} \tag{1}$$

where $SS_{xx}$, $SS_{yy}$, $SS_{xy}$, $SS_{zz}$, $SS_{yx}$, $SS_{yz}$, $SS_{zy}$, $SS_{zx}$, and $SS_{xz}$ are the summation of squares between two variables. The summations of squares were taken from the following matrix, which was developed using MATLAB script to calculate the Pearson correlation coefficients. The summation of squared matrix shows the dispersion of data and mean values of the indicators, which are described in following the sum of squared matrix (SSM):

$$= \frac{1}{(n-1)} \times \begin{bmatrix} SS_{xx} & SS_{xy} & SS_{xz} \\ SS_{yx} & SS_{yy} & SS_{yz} \\ SS_{zx} & SS_{zy} & SS_{zz} \end{bmatrix}$$

where, n = N × N Matrix value,

$$SS_{xx} = \sum (xi - \overline{x})^2 \text{ and}$$

$$SS_{xy} = \sum (xi - \overline{x}) \times (yi - \overline{y}).$$

similarly,

$$SS_{yz} = \sum (y_i - \overline{y}) \times (z_i - \overline{z})$$

where, $xi$, $yi$, $zi$ = actual variables at the respective column x, y, and z; and $SS_{xx}$, $SS_{xy}$, and $SS_{yz}$ represent the deviations predicted from actual values of $xi$, $yi$, and $zi$, respectively. It is obvious that the correlation interprets the relationship between two variables whereas the correlation matrix suggests the dependency between the variables. For the formation of the correlation matrix, sum of squared data is essential.

Furthermore, $\overline{x}$, $\overline{y}$, $\overline{z}$ = mean value of the respective column x, y, and z.

Pearson correlation coefficients were used to develop the correlation matrix that helped to find the matrix of relative weights of indicators, where this matrix of relative weights reflected the Eigenvector [13], which came from the largest Eigenvalue [14]. Eigenvalues are a special set of scalars associated with a linear system of equations (i.e., a matrix equation) that are sometimes also known as characteristic roots, characteristic values, proper values, or latent roots [15]. The relative weight [16] matrix is highly dependent on the interrelations among the indicators as it expresses the true statistical interdependence [17]. Using Equation (1), the correlation matrix was established, which is described as

$$= \begin{bmatrix} 1 & Pxy & Pxz \\ Pyx & 1 & Pyz \\ Pzx & Pzy & 1 \end{bmatrix}.$$

Using the above relation, a set of Eigenvalues was formed and with the largest Eigenvalue, a set of Eigenvectors was generated, which reflected the desired weights of indicators through the following sequences of Equations (2) to (4). This set of vectors gave the relative weights among the indicators.

To find the Eigenvector, the following equations were applied, which are represented by:

$$[A] \times [w] = \lambda\text{max} \times [w];$$    (2)

where, $A$ is the correlation matrix defined in relation (3), which represents the relation between the indicators, and $w$ is the weight vector, which is mainly the Eigenvector for the largest Eigenvalue $\lambda_{max}$.

To solve the Equation (2), $\lambda$ is needed to find solution of Det (A–λI) = 0. So,

$$\left\| \begin{bmatrix} 1 & x & y \\ x & 1 & z \\ y & z & 1 \end{bmatrix} - \lambda \times \begin{bmatrix} 1 & 0 & 0 \\ 0 & 1 & 0 \\ 0 & 0 & 1 \end{bmatrix} \right\| = 0$$    (3)

After solving the above Equation (3), a set of Eigenvalues is formed. The desired Eigenvector corresponding to the largest Eigenvalue indicates the relative weights and it is expressed as:

$$\begin{bmatrix} \lambda 1 \\ \lambda 2 \\ \lambda 3 \end{bmatrix} \quad \text{Largest Eigenvalue} \quad \begin{bmatrix} w1 \\ w2 \\ w3 \end{bmatrix}$$

Eigenvalues                    Eigenvector    (4)

A MATLAB script (see in Appendix A) is used to analyze the Eigenvalues associated with Eigenvectors.

Apparently, the weighting approaches of MSF and PCA look similar. In both methods, Eigenvector are characterized, and the covariance matrix does nearly the same measurement. This can also be determined by correlation matrix with sum of squared deviation.

PCA results find two or more components, which are merely retained with Eigenvalues greater than 1 [18]. Each component has variance, which describes how one component vector deals with the variables. The variances represent the weights of component vectors. Then the expected relative weights of indicators are calculated from the linear combination of component vectors with the variances (weights of component vectors), which is complex, whereas MSF directly gives the 'decision' to select the Eigenvector as relative weights that correspond to the largest Eigenvalue. For the clarification of taking Eigenvalue, an approach [19,20] is considered, which states that, "the set of weights is the Eigenvector, which gives the largest Eigenvalue". In terms of an application to assign weights for a large number of indicators, MSF is comparatively easy and simple to apply compared to PCA. It does not need any 'decision' to be taken about the 'components'.

## 5. Application of MSF for Vulnerability and Risk Assessment of the Bangladesh Coast

Large numbers of past studies are available in the literature where different weight computation methods are applied to assess vulnerability of the Bangladesh coast. Roy assessed spatial vulnerability of floods in the coastal regions of Bangladesh by using the analytical hierarchal process (AHP) [21]. Ahsan assessed socio-economic vulnerability due to climate change by using the expert opinion method for giving weight to the indicators in the south-western coastal region of Bangladesh [22]. Younus assessed vulnerability to cyclones for a village named Bawalkor in the Barguna district in Bangladesh by using the PRA survey method for giving weights to the indicators [23,24]. Kazi Ali Toufique assessed vulnerability of livelihoods in the coastal districts of Bangladesh by using the equal weightage method [25]. Aminul Islam constructed an area-based climate and disaster vulnerability index for Bangladesh by using principal component analysis (PCA) for giving weight to the indicators [26].

In this study, MSF was applied in a real-life situation to assess the degree of 'harm' due to a specific 'hazard'. Due to a cyclone generated storm surge hazard, as an application of MSF, socio-economic vulnerability and risk ('harm') [27] in the Bangladesh coast ('system') was assessed. An index based approach [28] of vulnerability was considered by selecting two major domains with a total of 15 indicators (Table 1). Five of them were social index indicators, three were economic index indicators, and the rest were disaster bearing capability index indicators. The social index indicators were number of population, number of household, population density, male–female ratio, and social dependence. Economic index indicators were poverty rate, type of household, and road grade. Disaster bearing capability index indicators were divided into structural and non-structural types. Structural types were water supply, cyclone shelter, polder embankment, and road density. Non-structural types were education level, drinking water availability, and labor ratio. Socio-economic data from the recent census were collected from the Bangladesh Bureau of Statistics (BBS) [29].

All indicators are not of equal importance. Therefore, proper contribution of indicators is needed. In this case, MSF was applied to give actual importance to the indicators.

Figure 1 describes the flowchart with selected indicators and the respective domains for the socio-economic vulnerability assessment for storm surge hazard.

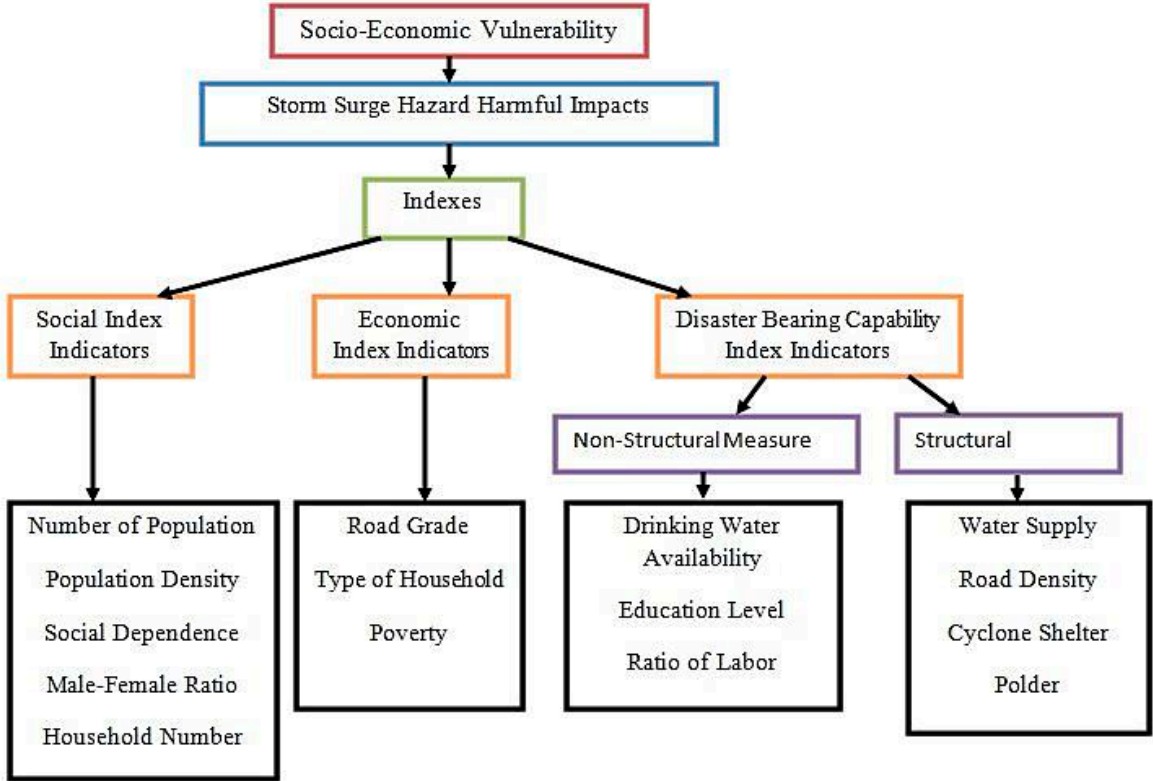

**Figure 1.** Flowchart of socio-economic vulnerability assessment for storm surge hazard.

The development of socio-economic (Griggs et. al., 2002; Turner et al., 2003; Ranganathan, 2009) vulnerability indexes adopts weights as shown in Equation (5), where the weight of each indicator is multiplied to the scaled score for each indicator:

$$\text{Index} = \sum (\text{Indicator} \times \text{Weight}) \qquad (5)$$

MSF was applied to compute the weights required by Equation (5). Three indexes were considered, where each index consisted of several indicators (Table 1). The weights of the indicators were computed by applying MSF. Table 2 lists the largest Eigenvalue of indexes that were used to determine the weights of each of the indicators.

**Table 1.** Vulnerability indicators.

| Domain | Socio-Economic Indicators | Indicator Description | Data Description |
|---|---|---|---|
| Social Index Indicators | Number of population | Populations is susceptible to hazard impacts [30]. | Data source: BBS, 2011. Data unit: Total number of population in each upazila. |
| | Number of households | Increased number of households leads to increased risk [31,32] | Data source: BBS, 2011. Data unit: Percentage of number of households per unit upazila area (square kilometer). |
| | Population density | Denser population increases risk due to lack of quality housing and social services network, which may not have had time to adjust with the population demand [32,33] | Data source: BBS, 2011. Data unit: Number of population per unit upazila area (square kilometer). |
| | Female to male ratio | Female to male ratio has impact on vulnerability; hence it increases risk as female population are more vulnerable than male population. | Data source: BBS, 2011. Data unit: ratio of female to male population |
| | Social dependence | Children and elders are the most vulnerable groups in hazards [34] Children, especially in the youngest age groups, cannot protect themselves during disasters like storm surges [35] because they lack the necessary resources, knowledge, or life experiences to effectively cope with the situation [36]. | Data source: BBS, 2011. Data unit: Percentage of summation of women, children under the age of 18, and elderly people to the total upazila population. |
| Economic Index Indicators | Poverty rate | Poverty rate has serious impact on vulnerability. Higher poverty rate results in higher vulnerability. Poverty level generally indicates the social status, standard, and dignity [37]. | Data source: World bank report, 2010. Data unit: Percentage of extreme poor lying below poverty line. |
| | Type of household | A stronger house, like a packa and semi packa house, reduces risk, whereas a weak and unoccupied house, like a kutcha and jhupri house, amplify risk [25,32]. | Data source: BBS, 2011. Data unit: Percentage of Kutcha and Jhupri house per upazila. Here Kutcha and Jhupri means houses made with timber, log, and tree leaf. |
| | Road grade | Road grade indicates the various types and classes of roads. It symbolizes the economic condition in such a way, where there are various types and classes of roads, regions are very much capable of handling all types of facilities because of having transportability [38]. | Data source: BBS, 2011. Data unit: Percentage of metaled and semi-metaled road length to total length of road per upazila. |
| Disaster Bearing Capability Index Indicators (structural measures) | Water supply | Where there is a suitable water supply, higher road density, large number of cyclone shelter and embankments, the resilience will increase [39]. | Data source: BBS, 2011. Data unit: Percentage of tap water and other pond type surface water per upazila area |
| | Cyclone shelter | Cyclone shelter is a structural measure that increases resiliency of a community to cope up with the adverse consequences of storm surge hazard [32,40]. Shelters and households have direct impacts on storm surge hazard [34]. | Data source: CEGIS, 2009. Data unit: Number of cyclone shelters per upazila population. |
| | Polder embankment | Polder is a flood control embankment, which is considered as a structural adaptation to reduce flood risk [32,41]. | Data source: BWDB, 2012. Data unit: Percentage of total poldered area (km2) per upazila area. |
| | Road density | Significant amount of road density ensures improved mobility/accessibility to services. It increases coping capacity of a community in case of any hazard occurrence. | Data source: BBS. Data unit: Road length per upazila area. |
| Disaster Bearing Capability Index Indicators (non-structural measures) | Education level | Illiterate people are more vulnerable than literate people [25,31–33]. | Data source: BBS, 2011. Data unit: Percentage of number of literate people per upazila population. |
| | Drinking water availability | When higher percentage of households drink unsafe water (tap, pond, and other open water), risk is increased [25,31–33]. | Data Source: BBS, 2011. Data Unit: Percentage of safe drinking water source to total population per upazila population. |
| | Labor ratio | Employed populations are less vulnerable to climatic hazards as they have high capability to cope with the vulnerable situation [42]. | Data source: BBS, 2011. Data unit: Percentage of employed people to total population. |

**Table 2.** The largest Eigenvalue of indexes considered for storm surge vulnerability.

| Indexes | Largest Eigenvalue | |
|---|---|---|
| Social Index Indicators | 2.955 | |
| Economic Index Indicators | 1.355 | |
| Disaster Bearing Capability Index Indicators | Non-Structural Measure | Structural Measure |
| | 1.930 | 1.599 |

The relative weights (which were determined using MSF), as shown in Table 2, were applied for vulnerability and risk assessment along the Bangladesh coast. The IPCC A5 approach [3] was used to assess vulnerability and risk. This approach needed hazards to compute risk. The storm surge hazard map was prepared by using a model simulated surge depth due to cyclone Sidr that made landfall on the Bangladesh coast in 2007 [43]. To prepare the hazard map, model simulation was done by considering several types of Sidr-like cyclones to make landfall in different locations along the Bangladesh coast. During model simulation, polder (one of the disaster bearing capability index) was considered to intervene in surge propagation.

Figure 2 shows the socio-economic vulnerability and risk maps generated by applying MSF. The vulnerability and risk assessment by MSF showed relatively high vulnerability and risk due to storm surge hazards in the central and exposed eastern coast of Bangladesh.

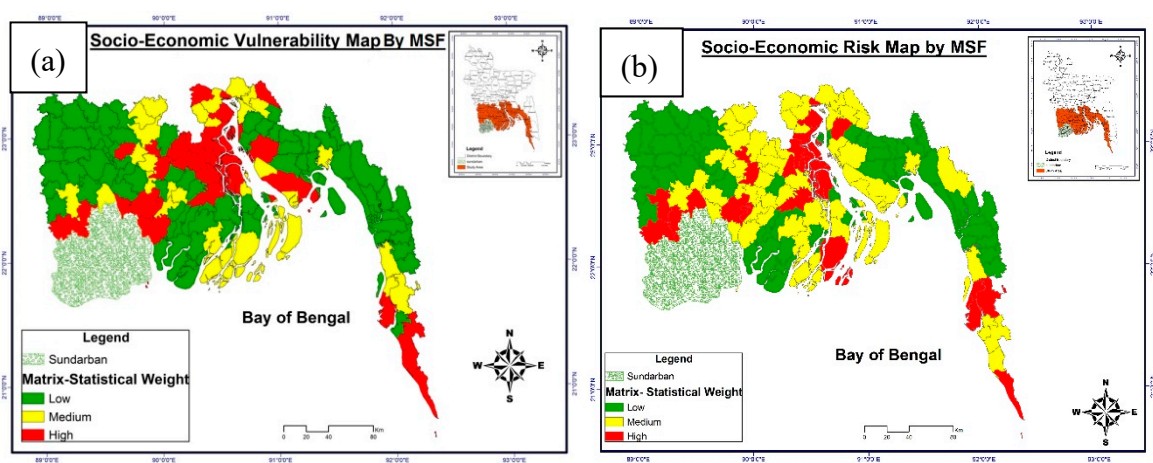

**Figure 2.** Vulnerability (**a**) and risk (**b**) maps using matrix based statistical framework (MSF).

## 6. Comparison of Weights Computed by MSF with Other Methods

Table 3 shows the weights computed for various socio-economic indicators by MSF along with other weight computation methods.

For social index, according to Table 3, household number gets the maximum weight from MSF. Physically, households are the very first element affected by storm surge and damages on households maximizes the storm surge risk. Chronologically the weight values computed by MSF were: population density, male–female ratio, number of population, and social dependence. The magnitude of weights for population density and male–female ratio were very close to the magnitude of weight for household number. The population in certain area with its male–female components was affected by the storm surge along with household number. For economic index, MSF assigned the maximum weight to household types followed by poverty. The household types symbolized the economic status of the population. The poverty rate was a direct measure of economic condition. Disaster bearing capability index defined the adaptations that lowered the vulnerability of storm surge hazard. The indicators considered were divided into structural and non-structural measures. Among the structural measures, MSF gave the maximum weights to water supply and road density; both were considered as lifelines

in hazard locations. Presence of these two lifelines lowered the vulnerability during storm surge. MSF correctly recognized the importance of these indicators by assigning maximum weights to them. Among the non-structural adaptation measures, labor ratio (number of working force available), and education level (number of educated people available) got the maximum weights from MSF. These two indicators represented the level of capability of society to counter a hazard, and the awareness of people. In this case also, MSF correctly recognized the importance of these two indicators and assigned maximum weights to them. Weights computed by other methods are also shown in Table 2. Among them, the equal weight method gave equal importance to all the indicators, which was physically not possible. Expert weighting was biased by individual thought and applicable in micro scale only. In some cases, for example, when weights of household number or types of household or education level were considered, experts gave judgments that were apparently compatible with MSF. However, the magnitude of these weights showed a clear bias towards a specific indicator.

**Table 3.** Weights of various socio-economic indicators computed by MSF and other weight methods.

| Socio-Economic Indicators | MSF | Explicit Weighting | | Statistical Weighting | |
|---|---|---|---|---|---|
| | | Equal Weighting | Expert Weighting | PCA | AHP |
| Social Index | | | | | |
| No. of Population | 16.42 | 20 | 15 | 13.65 | 9.84 |
| Population Density | 25.07 | 20 | 25 | 24.32 | 19.15 |
| Male–Female Ratio | 24.65 | 20 | 10 | 19.60 | 25.95 |
| Social Dependence | 8.47 | 20 | 20 | 15.16 | 25.87 |
| Household Number | 25.39 | 20 | 30 | 27.27 | 19.19 |
| Economic Index | | | | | |
| Type of Household | 48.34 | 33.33 | 50 | 33.33 | 70.40 |
| Road Grade | 3.85 | 33.33 | 20 | 29.34 | 0.32 |
| Poverty | 47.81 | 33.33 | 30 | 37.33 | 29.28 |
| Disaster Bearing Capability Index | | | | | |
| Structural Measure | | | | | |
| Water Supply | 39.84 | 25 | 10 | 37.33 | 14.25 |
| Road Density | 37.78 | 25 | 25 | 8.12 | 3.26 |
| Cyclone shelter | 11.80 | 25 | 30 | 12.57 | 7.76 |
| Polder | 10.58 | 25 | 35 | 41.98 | 74.73 |
| Non-Structural Measure | | | | | |
| Drinking Water Availability | 14.36 | 33.33 | 30 | 18.76 | 42.58 |
| Education Level | 42.16 | 33.33 | 50 | 40.21 | 33.53 |
| Labor Ratio | 43.48 | 33.33 | 20 | 41.03 | 23.89 |

PCA, on the other hand, selected the principal components, which were then used to compute the weights of indicators. As PCA calculated weights by using linear combination of component vectors with variances, it basically represented the weights of component vectors. Due to this, individual impact of indicators was not explicitly treated by PCA.

AHP, being a pair-wise comparison method, was not applicable when a large number of inter-dependent indicators was present. This was reflected in the computed weights where some of the indicators got abnormally high weight (for example polder in Table 3).

## 7. Accuracy of MSF Compared to Other Methods

In this section, accuracy of MSF was determined by comparing it with other methods. In doing so, socio-economic vulnerability maps were prepared by all the methods after applying the weights shown in Table 3.

The strategy taken here for comparison was to compare the storm surge hazard map with the vulnerability maps prepared by all methods including MSF. Vulnerability indicates the extent of harm and this can be measured by the indicators that have the direct impact by any climatic hazard. Without hazard, harm cannot happen. So, to compare accuracy of MSF, the socio-economic vulnerability was considered, which was triggered by the storm surge hazard. The storm surge hazard map describes the hazard intensity with spatial variation, whereas, the socio-economic vulnerability map shows the degree of damages for storm surges. The overlapping approach for these two maps formulated the strategy to measure accuracy of MSF compared to other methods. In the literature, this type of approach is known as a 'qualitative comparison' method [21], where hazard is considered as a 'prototype' and vulnerability is considered as a 'model'.

The comparison between the storm surge hazard map and vulnerability map prepared by MSF is shown in Figure 3. The visual comparison shows a similar pattern of hazard and socio-economic vulnerability and indicates the accuracy of MSF. The upper central coastal region was hazardous as well as vulnerable due to a storm surge (Figure 3).

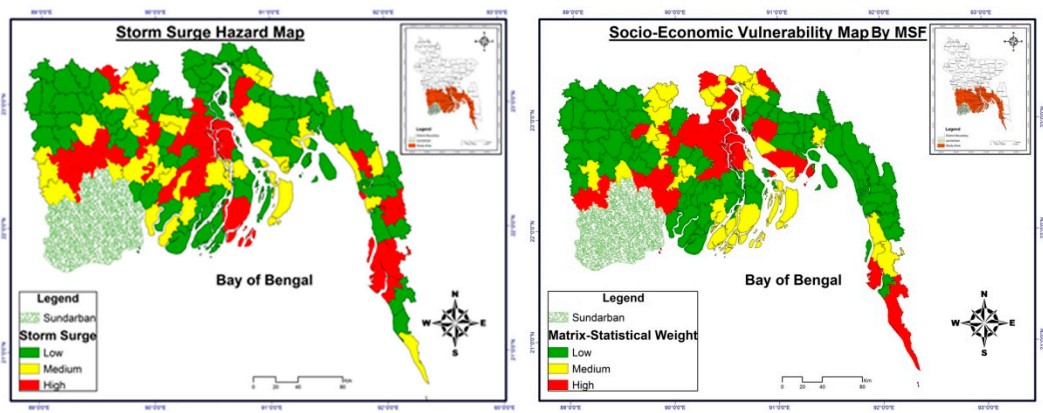

**Figure 3.** Comparison between storm surge hazard map (left) and socio-economic vulnerability map (right) prepared by using MSF.

To compare accuracy of MSF with other methods, comparisons of socio-economic vulnerability maps using the same set of indicators (Table 3) prepared by using equal weighting, expert weighting, PCA, and AHP with the storm surge hazard map are shown in Figure 4.

To translate the qualitative visual comparison into a quantitative one, one-to-one function was defined among the colors. When spatial distributions of colors were similar between any two maps, it was termed as 'exactly-similar'. In this way, two more categories were defined, namely 'partly-similar' and 'not-similar'. When several maps were compared with the same base map, the map that showed the largest percentage for the 'exactly-similar' category and the smallest percentage for the 'not-similar' category was considered to have the maximum similarity with the hazard map. Quantitative comparison [44] of all the vulnerabilities maps with the hazard map is shown in Table 4. It was found that the vulnerability map prepared by using the MSF had the largest percentage (49%) of the 'exactly-similar' category. Therefore, this map had the maximum similarity with the hazard map. As mentioned earlier, the hazard map was considered as 'prototype' and vulnerability map was considered as 'model'. With this assumption, the comparison showed that MSF performed the best among all other existing methods. Therefore, we can say that MSF is the most 'accurate' and simplest method among equal weighting, expert opinion, PCA, and AHP.

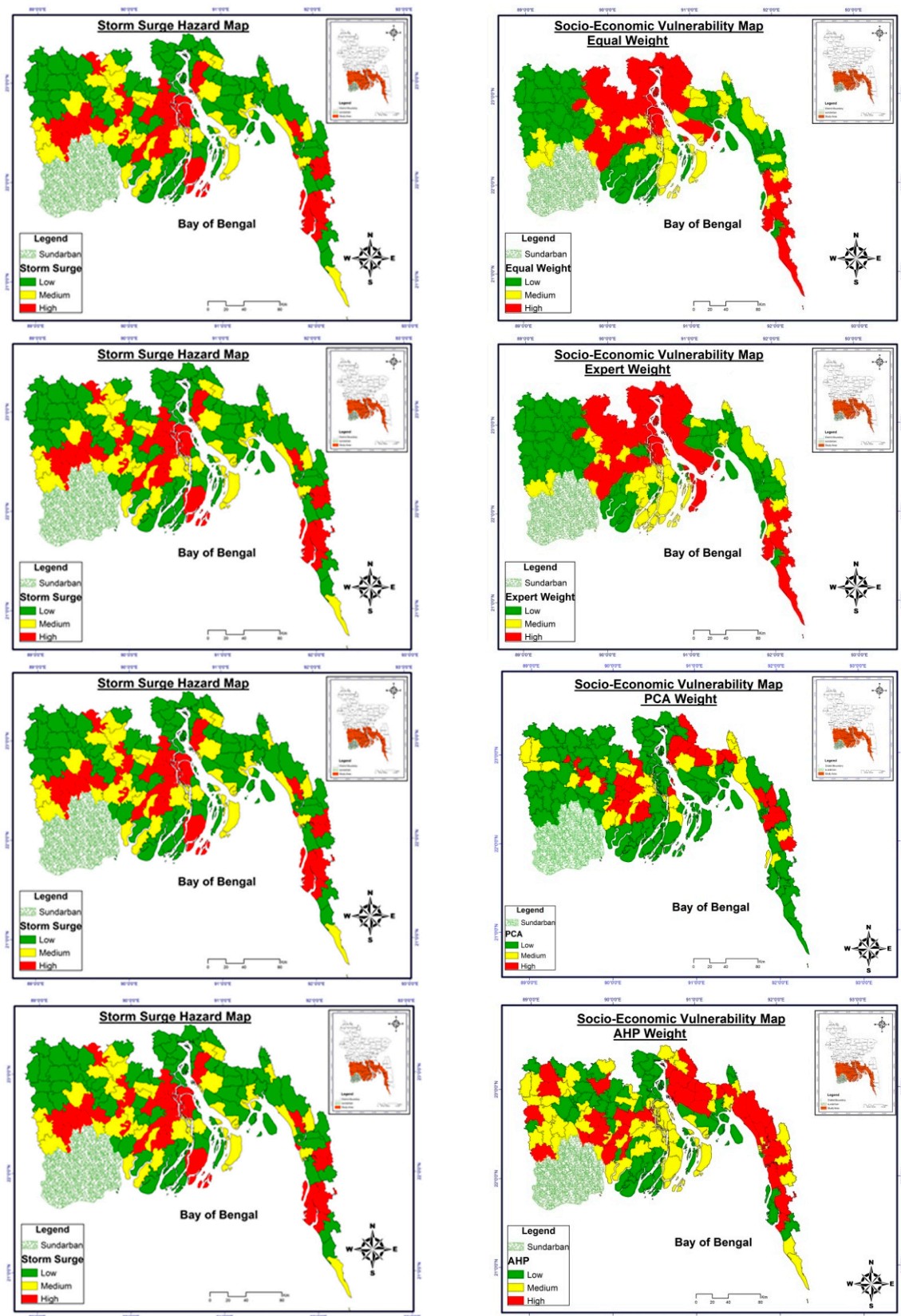

**Figure 4.** Comparison of storm surge hazard map (left) and socio-economic vulnerability maps (right) prepared by using (from top to bottom) equal weighting, expert weighting, PCA, and AHP.

**Table 4.** Accuracy of MSF compared to other weight methods.

| Weighting Method | Similarity in Percentage with Hazard Map | | |
|---|---|---|---|
| | Exactly-Similar | Partly-Similar | Not-Similar |
| MSF | 49 | 27 | 24 |
| Equal Weighting | 41 | 31 | 28 |
| Expert Opinion | 42 | 29 | 29 |
| PCA | 47 | 25 | 28 |
| AHP | 35 | 41 | 24 |

## 8. Conclusions and Recommendations

This paper introduces a new method to compute weights of indicators for composite hazard, vulnerability, and risk assessments. The method is named 'matrix based statistical framework' (MSF). To develop this method, valuation of correlation matrix and Eigenvector associated with Eigenvalue is considered from Pearson correlation coefficients. A MATLAB script is used to analyze Eigenvectors and Eigenvalues. To demonstrate the application of MSF, vulnerability, and risk along the Bangladesh coast is assessed by using MSF as the weighting method. A total of 15 socio-economic indicators are used to generate a vulnerability map, and storm surge hazard is considered to generate a risk map of the Bangladesh coast by following the IPCC AR5 approach, where weights to various indicators are assigned by MSF. The weights computed by MSF are compared with the weights computed by other methods for the same set of indicators. It is found that MSF gives the best physically acceptable explanation of the values of weights compared to the other methods. Accuracy of MSF is compared with the other weight methods by considering hazard as the prototype and vulnerability as the model. It is found that the vulnerability map prepared by using MSF has the maximum similarity (49%) with the prototype compared to other weight methods. MSF is a relatively simple statistical method, which directly gives the decision to select the Eigenvector as the relative weight that corresponds to the largest Eigenvalue. When the application involves a large number of indicators in any geographical setting, it is recommended to apply MSF as it is relatively accurate and easy to apply compared to other weight computation methods.

**Author Contributions:** Conceptualization, R.K. and D.S.K.; Data curation, M.S.; Methodology, M.A.; Resources, M.R.; Supervision, A.H.

**Funding:** This research was funded by the DECCMA project that was carried out under the Collaborative Adaptation Research Initiative in Africa and Asia (CARIAA), with financial support from the UK Government's Department for International Development (DFID) and the International Development Research Centre (IDRC), Canada.

**Acknowledgments:** This work is a part of Master's thesis, which is supported by the DECCMA project that carried out under the Collaborative Adaptation Research Initiative in Africa and Asia (CARIAA), with financial support from the UK Government's Department for International Development (DFID) and the International Development Research Centre (IDRC), Canada. The views expressed in this work are those of the creators and do not necessarily represent those of DFID and IDRC or its Board of Governors. Website: www.deccma.com.

**Conflicts of Interest:** The authors declare no conflicts of interest.

## Appendix A

**Pseudocode**
```
[Val, Ind] = max(eig(corr(data)));
[EgVc, EgVl] = eig(corr(data));

Dim = size(data,2);

RowD = [];
```

```
for i = 1: Dim
    ColD = [];
    for j = 1: Dim
        t1 = data(:,i)-mean(data(:,i));
        t2 = data(:,j)-mean(data(:,j));
        ColD = [ColD,sum(t1.*t2)];
    end

    RowD = [RowD;ColD];
end

fprintf('SS = \n');
disp(RowD);

fprintf('Corr = \n');
disp(corr(data));

fprintf('Eigen Value = %f\n\n',Val);
fprintf('Eigen Vactor =\n');
disp(EgVc(:,Ind))
```

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
