# Peer review of "Development of a Matrix Based Statistical Framework to Compute Weight for Composite Hazards, Vulnerability and Risk Assessments"

_climate, doi:10.3390/cli7040056_

Round 1
Reviewer 1 Report
Title: Development of a Matrix based Statistical Framework to compute weight for Composite Hazards and Risk Assessment Reports
Comments: It is a good article and well written. The authors have focused on computing weighting methods through the MSF in comparison with the other methods - which can be a good methodological contribution in / for assessing vulnerability and adaptation in localized base level study area/s. The present structures do not reach up to that level which cannot be published in the international reputed journals, such as, Climate. I would make some suggestions – these should be added in the revised version of article – then it might be published in the journal: Climate; my suggestions are:
1. Where is the significance of this paper at the abstract section and also at the end of the introduction section;
2. Where are the applications and recommendations of the Matrix based Statistical Framework (MSF).
3. Where is the key objective (s) of this paper at the abstract section as well as at the introduction section; the term ‘key notes’ is not the enough indication for identifying the objective of this paper; The objective would be more clearer in this way: to ………in order to………….
4. The Title can be changed; such as,…… focusing on or a case of Bangladesh coast???
5. International literature regarding composite hazards and risk assessment relating to coast, particularly Bangladesh coast, have largely been missing;
6. Where is the participatory Rapid Appraisal and its link with weighting index; it is obvious that this has real-world implication/s for assessing local vulnerability. The process of participatory rapid appraisal and its weighting index has lots of advantages and disadvantages factors, these should be noted in this article as well. Please see further literature; you may consider some references below in order to identify and compare with the PRA / participatory rapid appraisals and the MSF;
a)Younus, M and Kabir, M. (2018): Climate Change Vulnerability Assessment and Adaptation of Bangladesh: Mechanisms, Notions and Solutions, Sustainability 2018, 10 (11), 4286; https://doi.org/10.3390/su10114286 MDPI, Switzerland. b)Younus, M (2017): An assessment of vulnerability and adaptation to cyclones through impact assessment guidelines: a bottom-up case study from Bangladesh coast, Natural Hazards, December 2017, Volume 89, Issue 3, pp 1437–1459. Springer. DOI 10.1007/s11069-017-3027-8 https://doi.org/10.1007/s11069-017-3027-8c)Younus, M (2017): Adapting to Climate Change in the Coastal Regions of Bangladesh: Proposal for the Formation of Community-Based Adaptation Committees, Environmental Hazards, 16:1, 21-49, http://dx.doi.org/10.1080/17477891.2016.1211984. d)Younus, M. (2014): Flood vulnerability and adaptation to climate change in Bangladesh: A Review, Journal of Environmental Assessment Policy and Management, Vol. 16, No. 3 (September 2014) 1450024 (28 pages), Imperial College Press, DOI: 10.1142/S1464333214500240, Print ISSN: 1464-3332, Online ISSN: 1757-5605.7. Is the MSF a new method? Why this method will use for assessing vulnerability and adaptation as a bottom up and localized impact assessment under the climate change regimes? What are the advantages and disadvantages of the MSF? These should be highlighted in a separate section not in the other texts;
8. Your conclusions and recommendation sections should be specific and pin-pointed, and this section needs to be little bit elaborated.
9. How much reliable of the method of MSF in comparison with the PRA along with other participatory approaches including Focus group discussion etc;
10. The relevant literature review section should separately be added focusing on the PCA, expert weighting, AHP, other weighting approaches/methods;
11. The Table 2 should clearly be indicated about the weights of vulnerability indicators, e.g. type of household means?? These variables should be clearly and meaningfully described in an index under the table;
12. Have you found out the differences between your proposed method and other method’s level of significance or co-efficient correlations under each variable? What are the differences of these relationships from the standard deviation or their skewness?
Author Response
Reviewer #2: Manuscript ID: climate-445057
Title: Development of a Matrix based Statistical Framework to compute weight for Composite Hazards and Risk Assessment Reports
1. Include the findings of this study in the abstract
Answer: In the revised manuscript, abstract is re-written where findings of the study are included
2. The introduction needs to identify what the contribution of this manuscript to the broader literature given that vulnerability and risk assessment has been addressed in many disciplines
Answer: In the revised manuscript, introduction section is re-written by highlighting contribution of the paper to the broader literature in the context of vulnerability and risk assessment
3. The introduction also needs to identify why this study is needed in the context of the literature.
Answer: In the revised manuscript, this specific aspect is mentioned.
4. The background section discusses weighting methods that are generally used for these kinds of studies. However, this section also needs to provide a significant discussion of the studies undertaken in this area before concluding the need for this study.
Answer: During description of each of the methods in background section, gap and limitations of the method is addressed that ultimately concludes why a new method is needed. This is also addressed in the introduction section.
5. The conclusion section is sparse. The authors need to discuss what is the advantage of the proposed method from theoretical and practical perspective, and also from operational perspective. How should someone interpret the results when using two different methods? Although the authors implemented a spatial similarity approach, how does the variation based on weight of indicator influences interpretation of the results from stakeholder perspective?
Answer: In the revised manuscript, the conclusions are re-written.
A separate section (section 7 in the revised manuscript) is added that describes advantages and disadvantages of the new method.
In section 5 of the revised manuscript, physical significance of the weights computed by MSF compared to other weight methods are explained.
6. The authors also need to provide the pseudo code they implemented to help this study be reproducible.
Answer: Pseudo code is included as an Appendix in the revised manuscript.

Reviewer 2 Report
Title: Development of a Matrix based Statistical Framework to compute weight for Composite Hazards and Risk Assessment Reports
Comments: It is a good article and well written. The authors have focused on computing weighting methods through the MSF in comparison with the other methods - which can be a good methodological contribution in / for assessing vulnerability and adaptation in localized base level study area/s. The present structures do not reach up to that level which cannot be published in the international reputed journals, such as, Climate. I would make some suggestions – these should be added in the revised version of article – then it might be published in the journal: Climate; my suggestions are:
1. Where is the significance of this paper at the abstract section and also at the end of the introduction section;
2. Where are the applications and recommendations of the Matrix based Statistical Framework (MSF).
3. Where is the key objective (s) of this paper at the abstract section as well as at the introduction section; the term ‘key notes’ is not the enough indication for identifying the objective of this paper; The objective would be more clearer in this way: to ………in order to………….
4. The Title can be changed; such as,…… focusing on or a case of Bangladesh coast???
5. International literature regarding composite hazards and risk assessment relating to coast, particularly Bangladesh coast, have largely been missing;
6. Where is the participatory Rapid Appraisal and its link with weighting index; it is obvious that this has real-world implication/s for assessing local vulnerability. The process of participatory rapid appraisal and its weighting index has lots of advantages and disadvantages factors, these should be noted in this article as well. Please see further literature; you may consider some references below in order to identify and compare with the PRA / participatory rapid appraisals and the MSF;
a)Younus, M and Kabir, M. (2018): Climate Change Vulnerability Assessment and Adaptation of Bangladesh: Mechanisms, Notions and Solutions, Sustainability 2018, 10 (11), 4286; https://doi.org/10.3390/su10114286 MDPI, Switzerland.
b)Younus, M (2017): An assessment of vulnerability and adaptation to cyclones through impact assessment guidelines: a bottom-up case study from Bangladesh coast, Natural Hazards, December 2017, Volume 89, Issue 3, pp 1437–1459. Springer. DOI 10.1007/s11069-017-3027-8 https://doi.org/10.1007/s11069-017-3027-8c)Younus, M (2017): Adapting to Climate Change in the Coastal Regions of Bangladesh: Proposal for the Formation of Community-Based Adaptation Committees, Environmental Hazards, 16:1, 21-49, http://dx.doi.org/10.1080/17477891.2016.1211984.
d)Younus, M. (2014): Flood vulnerability and adaptation to climate change in Bangladesh: A Review, Journal of Environmental Assessment Policy and Management, Vol. 16, No. 3 (September 2014) 1450024 (28 pages), Imperial College Press, DOI: 10.1142/S1464333214500240, Print ISSN: 1464-3332, Online ISSN: 1757-5605.
7. Is the MSF a new method? Why this method will use for assessing vulnerability and adaptation as a bottom up and localized impact assessment under the climate change regimes? What are the advantages and disadvantages of the MSF? These should be highlighted in a separate section not in the other texts;
8. Your conclusions and recommendation sections should be specific and pin-pointed, and this section needs to be little bit elaborated.
9. How much reliable of the method of MSF in comparison with the PRA along with other participatory approaches including Focus group discussion etc;
10. The relevant literature review section should separately be added focusing on the PCA, expert weighting, AHP, other weighting approaches/methods;
11. The Table 2 should clearly be indicated about the weights of vulnerability indicators, e.g. type of household means?? These variables should be clearly and meaningfully described in an index under the table;
12. Have
you found out the differences between your proposed method and other
method’s level of significance or co-efficient correlations under each
variable? What are the differences of these relationships from the
standard deviation or their skewness?
Author Response
please see the file attached

Round 2
Reviewer 1 Report
I am not convinced with the revised version. In fact the authors have not substantially changed as per the comments which I made last time. I mentioned some key things, such as: significance of the article, and how to write the objective etc. Please carefully follow the comments and respond as per the opinions with relevant information. It seems to me it is still a Master's thesis or part of a thesis. An international article is different than the writing style of a thesis. You have to carefully write the conceptual setting of the article which is critical. The research question, research gap, objective and aim, method, significance, conceptual setting, case study area context and why it is chosen, using variables and their outcomes, discussion, findings, solutions, limitations and finally conclusions and recommendations - these are the essential ingredients for an article. Could you please highlight the above texts in the article and send it to me again. Have you checked the cross references which I suggested before? The scientists who worked on the Bangladesh coast such as Adger, Smit, Younus, Younus and Harvey, Ahmad and Warrick, Brammer, Huq (coauthor/s), Shaw, Paul, B., Barua, Ahmed, Nishat etc - these references need to be added in the literature review section. I will check your methodology section as well, later. It is obvious I need to know how you weight and quantify each variable/s in the context of your case study area. You need to make strong arguments behind your validation and weighting.
Author Response
Question: I am not convinced with the revised version. In fact the authors have not substantially changed as per the comments which I made last time. I mentioned some key things, such as: significance of the article, and how to write the objective etc. Please carefully follow the comments and respond as per the opinions with relevant information. It seems to me it is still a Master's thesis or part of a thesis. An international article is different than the writing style of a thesis. You have to carefully write the conceptual setting of the article which is critical. The research question, research gap, objective and aim, method, significance, conceptual setting, case study area context and why it is chosen, using variables and their outcomes, discussion, findings, solutions, limitations and finally conclusions and recommendations - these are the essential ingredients for an article. Could you please highlight the above texts in the article and send it to me again. Have you checked the cross references which I suggested before? The scientists who worked on the Bangladesh coast such as Adger, Smit, Younus, Younus and Harvey, Ahmad and Warrick, Brammer, Huq (coauthor/s), Shaw, Paul, B., Barua, Ahmed, Nishat etc - these references need to be added in the literature review section. I will check your methodology section as well, later. It is obvious I need to know how you weight and quantify each variable/s in the context of your case study area. You need to make strong arguments behind your validation and weighting.
Reply:
In revised manuscript, following sub-sections are added within the section ‘Introduction and Statement of Problem’:
Research Gap, Research question, Objectives and Significance
Overview of other Weightage Methods and a Comparison with the new MSF Method
In the revised manuscript, few changes are also made to re-organize the paper. For example, the sections that contains ‘Advantages and Disadvantages of MSF’ is now shifted just after the section ‘Introduction and Statement of Problem.’ ‘Conclusions’ section is re-written as ‘Conclusions and Recommendations.’ With these addition and re-organization, conceptual setting of the article is re-designed in the revised manuscript.
The suggested references are tried to incorporate as much as possible as stated below:
Suggested Reference | Authors response |
Adger | Relevant reference is included |
Smit | Reference is found on vulnerability and adaptation issue in Bangladesh coast, but no relevant reference is found related to weighting method. |
Younus | Relevant reference is included. |
Younus and Harvey | Relevant reference is included. |
Ahmad and Warrick | There are works on Bangladesh coast. But no relevant reference is found on weighting method. |
Brammer | There are works on Bangladesh coast. But no relevant reference is found on weighting method. |
Haque | There are works on Bangladesh coast. But no relevant reference is found on weighting method. |
Shaw-Paul, B | There are works on Bangladesh coast. But no relevant reference is found on weighting method. |
Barua | There are works on Bangladesh coast. But no relevant reference is found on weighting method. |
Ahmed | There are works on Bangladesh coast. But no relevant reference on weighting method is found. |
Nishat | There are works on Bangladesh coast. But no relevant reference on weighting method is found. |